# Health System- and Patient-Related Factors Associated with COVID-19 Mortality among Hospitalized Patients in Limpopo Province of South Africa’s Public Hospitals

**DOI:** 10.3390/healthcare10071338

**Published:** 2022-07-19

**Authors:** Musa E. Sono-Setati, Peter M. Mphekgwana, Linneth N. Mabila, Masenyani O. Mbombi, Livhuwani Muthelo, Sogo F. Matlala, Takalani G. Tshitangano, Naledzani J. Ramalivhana

**Affiliations:** 1Department of Public Health Medicine, University of Limpopo, Private Bag X1106, Sovenga, Polokwane 0727, South Africa; musasono@yahoo.com; 2Limpopo Department of Health, College Ave, Hospital Park, Polokwane 0699, South Africa; drramalivhana.c99986@gmail.com; 3Research Administration and Development, University of Limpopo, Private Bag X1106, Sovenga, Polokwane 0727, South Africa; 4Department of Pharmacy, University of Limpopo, Private Bag X1106, Sovenga, Polokwane 0727, South Africa; nkateko.mabila@ul.ac.za; 5Department of Nursing, University of Limpopo, Private Bag X1106, Sovenga, Polokwane 0727, South Africa; masenyani.mbombi@ul.ac.za (M.O.M.); livhuwani.muthelo@ul.ac.za (L.M.); 6Department of Public Health, University of Limpopo, Sovenga, Polokwane 0727, South Africa; france.matlala@ul.ac.za; 7Department of Public Health, University of Venda, Thohoyandou 0950, South Africa; takalani.tshitangano@univen.ac.za

**Keywords:** COVID-19, mortality, hospitalized, health system, patient-related factors

## Abstract

South Africa has recorded the highest COVID-19 morbidity and mortality compared to other African regions. Several authors have linked the least amount of death in African countries with under-reporting due to poor health systems and patients’ health-seeking behaviors, making the use of clinical audits more relevant for establishing the root causes of health problems, and improving quality patient care outcomes. Clinical audits, such as mortality audits, have a significant role in improving quality health care services, but very little is documented about the outcomes of the audits. Therefore, the study sought to determine the health care system and patient-related factors associated with COVID-19 mortality by reviewing the COVID-19 inpatient mortality audit narration reports. This was a retrospective qualitative research approach of all hospitalized COVID-19 patients, resulting in death between the first and second COVID-19 pandemic waves. Thematic analysis employed inductive coding to identify themes from mortality audits from all 41 public hospitals in Limpopo Province, South Africa. Four themes with seventeen sub-themes emerged: sub-standard emergency medical care provided, referral system inefficiencies contributed to delays in access to health care services, the advanced age of patients with known and unknown comorbidities, and poor management of medical supplies and equipment, as a health system and patient-related factors that contributed to the high mortality of COVID-19 patients. There is a need to routinely conduct clinical audits to identify clinical challenges and make recommendations for health promotion, risk communication, and community engagement. We recommend reviewing and expanding the scope of practice for health-care providers during epidemics and pandemics that include aspects such as task-shifting.

## 1. Introduction

The COVID-19 pandemic resulted in increased morbidity and mortality which overwhelmed and strained health systems in all countries. Globally, 6 million COVID-19-related deaths have been recorded, as of 15 March 2021. The African continent has recorded the least COVID-19 mortality, with around 170,942 deaths [1]. Several authors have linked the least amount of death in African countries with under-reporting due to poor health systems and patients’ health-seeking behaviors [2,3], making the use of clinical audit more relevant. South Africa has recorded the highest morbidity and mortality compared to other African regions [1,4]. Given these statistics, it is important to conduct a clinical audit to establish root causes of health problems and improve the quality of patient care. Historically, clinical audits have been recommended as a quality improvement tools. For instance, in the mid 1800s, Florence Nightingale utilized clinical audits for wounded soldiers in a surgical unit to establish the root cause of high mortality rates [5]. She implemented guidelines for effecting a series of changes which were identified during the clinical audit, which resulted in a mortality rate that decreased from 40 to 2%.

According to Young et al. (2011), a clinical audit is one of the best methods that health care organizations can use to understand and ensure the quality of services delivered [6]. Tantrige (2014) indicate that clinical audits help to identify gaps during patient care to improve the quality of care [7]. Moreover, clinical audits can inform the development of policies and guidelines based on identified gaps in clinical practice, environment and system failure. More importantly, interventions driven from clinical audits empower health care professionals to provide quality health care services [7]. A clinical audit provides an opportunity for health care professionals to share ideas and use problem-solving skills that can improve the current system of health care in South Africa.

All health systems want to provide quality health care services to users. Lapses in patient safety have been listed as one of the major problems in health care quality. In the United States of America, the third leading cause of death in 2016 was reported as health system errors and complications [8]. The most common types of preventable health system errors were those in diagnosis, delayed diagnosis, medication errors, and deficient medical devices [9]. Medication errors were reported to be common among COVID-19 patients in a large tertiary care setting in the Kingdom of Saudi Arabia. They were caused by dosing errors and errors in drug frequency, primarily attributed to physicians, which may indicate burnout or stress among them [10]. Most causes of death were documented on the death certificate; however, medical error did not seem to appear as the cause of death on death certificates [8], which is a health care system challenge.

Various authors have reported important challenges and recommendations in the health system during the COVID-19 pandemic [11,12]. For example, health system errors were identified as possibly the most critical challenge facing health care services [11,12]. Health system errors are common in intensive care units (ICU), operating rooms, and emergency departments (EDs) due to various factors [11,13]. The current lack of definitive treatment for COVID-19 has led health care providers to rely on uncertainties when managing patients, and this has negative consequences on the health system [1,14]. In addition, health system errors, such as wrong drug prescription, dosage, and injection, endanger the safety of patients [11]. Moreover, other errors, such as misdiagnosis or surgical mishaps resulting in treatment delays, worsened outcomes, such as fatal consequences or prolonged care, which jeopardize patient safety and breach public trust [15,16,17]. Patients are also contributors to medical error or patient safety. Lack of knowledge about medication and non-adherence to medical recommendations are some patient-related factors [18]. Consequently, health system errors are not only public or human health issues but increase the overall cost of hospitalization, and the legal costs associated with lawsuits, and eventually, decrease the quality of the health care system [10]. 

According to the National Policy for Patient Safety Incident Reporting and Learning (NPSIRL) in the Public Health Sector of South Africa (PHSSA), health system errors should be revealed to patients and their families [19,20]. However, about 10 to 20 percent of errors are never reported. This might be due to a lack of disclosure of medical error policies in health care institutions, fear of punishment and job loss amongst health care workers, as well as fear of a lawsuit by the health department [21,22]. In this regard, Esposito and Dal Canton emphasized the importance of clinical audits to improve reporting of health system errors and in clinical governance to improve quality of care by identifying inconsistencies between standard and actual practice without blaming health care providers [23].

A recent study by Thomas et al. identified COVID-19-related challenges in health care facilities as a shortage of medical supplies [24] and delays in laboratory processing of specimens resulting in poor quality patient care [24]. Subsequently, the Limpopo Province Department of Health (LPDoH) recently reported R14 billion medical negligence claims as one of the outcomes of poor-quality patient care [25]. The current study sought to determine the health care system and patient-related factors by reviewing the COVID-19 inpatient mortality audit narration reports. The clinical audit findings will assist in improving the quality of patient care during COVID-19 management, safeguarding high standards and reducing the mortality rate in future outbreaks, and to provide an opportunity for health care providers to reflect on and improve clinical practices. Additionally, policymakers, health care planners, and researchers must determine the challenges and preventative strategies in health care system errors.

## 2. Materials and Methods

### 2.1. Study Setting, Design and Sampling

This study used secondary data on the COVID-19 response kept by the LPDoH. The mixed-methods study protocol for the use of secondary data kept by LPDoH is reported elsewhere [26]. A quantitative study by Tshitangano et al. describes, in detail, the process followed by LPDoH when collecting primary data on the COVID-19 response using a clinical mortality audit tool [27]. The study further describes the clinical mortality audit tool as having seven sections, being: facility details, demographic characteristics, clinical information, physical examination on admission, oxygen requirement during hospitalization, diagnosis on admission, and hospital outcome. The current study adopted a qualitative approach with a narrative design. It used narrative data extracted from the hospital outcome section of the clinical mortality audit tool. This section has three subsections on the cause of death, an evaluation of what happened, and proposed actions to prevent similar future problems. The evaluation of what happened subsection has space for doctors to write narrative notes describing the sequence of events that led to the death of the patient. It is these narrative notes that the current study extracted and analyzed to determine factors associated with deaths among COVID-19 hospitalized patients in Limpopo Province of South Africa.

### 2.2. Data Collection 

The LDoH created the clinical mortality audit tool for data collection, with extensive validation by Limpopo Department of Health officials as supported by the World Health Organization COVID-19 provincial support team. The clinical mortality audit tool was shared with each hospital audit team for administration purposes regarding the COVID-19 inpatient mortality cases. The mortality audit tool was piloted at Kgapane Hospital which included the Kgapane clinical manager, infection prevention and control (IPC) nurse, and DATCOV data capturer facilitated by World Health Organization (WHO) public health officials. Thereafter, it was adjusted, and the tool was administered and completed by selected facilitators identified by the Provincial Case Management and Audit team. The audit was conducted in a participatory manner, in a group setting with representatives from the hospital consisting of the hospital clinical manager, infection control nurse, and a DATCOV data capturer owing to their familiarity and knowledge of the context of the demise of COVID-19 inpatients. The clinical mortality audit tool was administered in a group setting under the supervision of a senior clinical manager and nursing service manager. In each hospital, the following members constituted the hospital audit team—see Table 1 below:

As shown in Table 1, hospital audit teams were asked to look over and prepare patient records in preparation for the audit, which was followed by a review of each file and a group discussion. The audit tool included open-ended question on “*Assessment and analysis to determine what happened (Describe the sequence of events leading to COVID-19 mortality)*”. In an excel spreadsheet, a DATCOV data capture recorded all the narrations of the mortality audit reports.

### 2.3. Data Analysis

Content analysis was conducted using a systematic coding and categorizing approach to explore data from 770 records within all 41 public hospitals to identify themes regarding health system and patient-related factors associated with COVID-19 mortality. The adoption of content analysis enabled authors to determine the trends and patterns of words used and their frequency and relationship [28]. The next step was to conduct open coding, create themes, and group the codes in higher order headings aligning with the objectives of the study themes, and then sub-themes were formulated as they emerged. Data were reassembled in new ways, and connections between codes and similar codes were grouped into broader themes and sub-themes. Two authors (MOM & L) completed the data analysis by immersing themselves in the data to obtain a sense of the whole transcripts before generating the themes and sub-themes. To validate the themes and sub-themes, four authors (PMM, LNM, MOM, & L) discussed and verified the generated themes and sub-themes, which were confirmed by the first author (MSS). Throughout the coding process, the authors met regularly via a virtual platform (Google meet) to discuss and finalize themes and sub-themes to improve data trustworthiness.

### 2.4. Ethical Clearance and Permission to Conduct the Study

Ethical clearance was obtained from Turfloop Research Ethics Committee (TREC/293/2021: IR) while gatekeeper permission to conduct the study was obtained from Limpopo Province Department of Health (Ref: LP_2021-11-017) as indicated in the protocol [27]. Confidentiality, anonymity, and privacy of individual facilities and health workers were protected.

## 3. Results

A total of 770 medical records kept by the provincial office across 41 hospitals were used for secondary data analysis. As reported in our previously published study [27], hypertension was common amongst hospitalized COVID-19 deaths (*n* = 586, 64%), followed by diabetes mellitus (*n* = 450, 52%), HIV/AIDS (*n* = 141, 19%), and obesity (*n* = 81, 12%). In terms of beds shortage, approximately 57% of COVID-19 patients did not have admission beds in the persons under investigation (PUI) ward followed by the general ward, as reported in our study that was previously published [27].

Table 2 below presents the themes and sub-themes from the thematic data analysis of the narrative data extracted from the clinical mortality audit tool. Three main themes are related to health system errors, and one theme relates to patient-related factors associated with COVID-19 mortality among hospitalized patients and are supported by narrative extracts from the clinical mortality audit tool.


*Theme 1: Sub-standard emergency medical care provided*


The clinical mortality audit indicated sub-standard emergency medical care provided to most COVID-19 patients in the emergency departments (ED). According to attorneys at O’Connor, Parsons, Lane and Noble [29], sub-standard emergency medical care refers to a failure by the health care providers to adhere to the appropriate standard of care when treating a patient. Standard emergency care for COVID-19 patients is significant in restoring the normal function of the respiratory or cardiovascular systems of the patient. The section below presents sub-themes relating to sub-standard emergency medical care provided to COVID-19 patients, as health system errors are prevalent within the province, shown in Table 2. 

Sub-theme 1.1: Delays in patient triage and clinical assessment

The study indicated delays in triage and clinical assessment of patients due to unprepared emergency trolleys and lack of essential key ED equipment. The extracts below from the clinical mortality audit tool support the findings:


*“Insufficiently prepared emergency trolley resulted in not regularly giving treatment.”*



*“Difficulty in performing a procedure due to inability to access equipment”.*



*“No high flow oxygen machine while patients saturation dropping”*


Sub-theme 1.2: Delays in laboratory turnaround time 

Laboratory investigations have a significant role in the medical emergency care of patients, including COVID-19 patients. The current study findings indicate a delay in receiving results from laboratories, resulting in delays in clinical decision-making. The extracts below support the findings:


*“Long turnaround time for laboratory results, or late receiving of PCR COVID-19 results.”*



*“Delay of the results from the laboratory”*



*“Delay from lab results”*


Sub-theme 1.3: Insufficient clinical and radiological investigations 

Some COVID-19 patients were not adequately investigated, as evidenced by the lack of X-ray results and vital signs not being consistently conducted. As a result, those patients were inadequately investigated and monitored. These results are supported by the following extracts from the mortality audit reports:


*“Poor monitoring by medical staff or monitoring was not done at all.”*



*“Inconsistent monitoring of vital signs.”*


Sub-theme 1.4: Insufficient active resuscitation of severe COVID-19 cases

Most COVID-19 patients died following a lack of active resuscitation in the ED. Active resuscitation is an emergency intervention provided by health care providers to acutely ill patients in the ED. The findings are supported by the following extracts from the mortality audit report.


*“Active resuscitation not performed.”*



*“Developed headache and later saturation deteriorated. Active resuscitation not done”.*


Sub-theme 1.5: Inadequate monitoring of COVID-19 in emergency unit and intensive care unit

Inadequate medical monitoring of patients contributed to poor outcomes for COVID-19 patients. Some COVID-19 patients who needed continuous monitoring in ED with a ventilator for respiratory conditions were neglected. Ventilated patients require admission into a high care or ICU ward after stabilization in ED. In the context of this study, inadequate medical monitoring is interpreted as a medical error by health care providers who failed to admit the COVID-19 patients to the ICU. The extracts below support the findings:


*“Patient needed intensive monitoring with a ventilator.”*



*“Patient presented with shortness of breath. The next level of care is when the patient deteriorates due to the lack of a ventilator in the ICU”.*



*Theme 2: Referral system inefficiencies contributed to delay in access to health care services*


Findings from the COVID-19 mortality audit report indicate the referral system’s inefficiencies that result from patient- or health care provider-related factors. The referral system for COVID-19 patients includes transferring patients from home to health care facilities and inter-facility transfers to higher levels of care for further management. The causes of delay in seeking health care are described in the two sub-themes below:

Sub-theme 2.1: Severe and critical cases presenting with hypoxia on admission

One referral system challenge noted from the mortality audit report includes COVID-19 patients seeking health care services already when in severe hypoxia. This is supported by the following: 


*“Late seeking of medical attention with severe respiratory distress.”*



*“Presented late with severe acute respiratory syndrome causing hypoxia.”*


The findings indicate that some COVID-19 patients with severe symptoms sought health services late from the hospital. We interpreted the late seeking of health care services as a patient-related error that intensified the mortality rate among hospitalized COVID-19 patients. The following extracts support these findings:


*“The patient saturation deteriorated to 37% while receiving oxygen.”*



*“The patient was identified as critically ill with a 65–70% saturation on double Oxygen.”*


Sub-theme 2.2: Patients presenting with respiratory complications on admission

The study findings indicate that most referred patients had respiratory diagnoses such as pneumonia and bronchial asthma. We interpret severe respiratory complications that impacted the COVID-19 mortality. The following extracts support the findings:


*“Severe COVID 19 pneumonia, the patient was oxygen-dependent with severe hypoxia.”*



*“Patient who died within 5 h presented with severe pneumonia—progressed rapidly.”*



*“Known asthmatic patient admitted with upper respiratory tract infection on treatment—condition deteriorated rapidly.”*



*Theme 3: Advanced age of patients with known and unknown comorbidities*


The review findings indicate that most patients who had suffered severe stages of COVID-19 were in their advanced age with known and unknown comorbidities. For example, in the audit report within Limpopo Province, some of the COVID-19 patients with known and newly diagnosed comorbidities, such as diabetic mellitus, kidney failure, and hypertension, had severe stages of COVID-19 [27]. The discussion below summarizes the sub-themes related to COVID-19 complications and comorbidities.

Sub-theme 3.1: Advanced age and respiratory complications

The COVID-19 mortality audit report indicates most COVID-19 patients in their advanced age died from respiratory tract complications. The following extracts support these findings:


*“A 64 years old, known HIV on treatment, admitted with lower respiratory tract infection and condition rapidly deteriorated.”*



*“COVID-19 Pneumonia, RVD, stayed on the ward while on rebreathe oxygen”.*


Sub-theme 3.2: Patients with diabetes mellitus, diabetic ketoacidosis (DKA), hyperglycemia, or hypoglycemia

The review findings indicate that most COVID-19 patients presented as diabetes mellitus (DM) patients with complications such as diabetic ketoacidosis (DKA), hypoglycemia, or hyperglycemia. This meant that diabetes mellitus was poorly controlled or was complicated by COVID-19 and its management. The following extracts support the findings:


*“93 y/o female, diabetes mellitus on treatment presented with persistent hypoglycemia and severe pneumonia. She demised within three days.”*



*“70 y/o female, Hpt & DM on Rx, presented with elevated HGT and severe pneumonia, she demised within 12 h.”*


Sub-theme 3.3: Advanced aged patients presented with comorbidities and severe stages of COVID-19

Our findings demonstrate COVID-19 patients with advanced age and comorbidities had a severe form of COVID-19. Most patients presented with poorly controlled and complicated non-communicable diseases such as hypertension. The following extracts support the findings:


*“69 year old, known with hypertension on treatment; admitted with congestive cardiac failure, diagnosed with COVID-19; condition deteriorated rapidly, and the patient demised.”*



*“70-year-old male known with hypertension, end-stage kidney disease, prostate cancer; admitted with a severe lower respiratory tract infection and uraemic encephalopathy.”*



*Theme 4:*
*Sub-standard management of COVID-19 cases, medical records and resources.*


The provision of health care services involves the management of health care resources such as medical supplies and materials. However, the current study findings indicate that inefficient medical supplies and equipment management contributed to the discussed mortality of COVID-19 patients. The sub-themes below discuss this theme further:

Sub-theme 4.1: Out of stock treatment in the wards and pharmacy department

Supply and management of medical supplies in COVID-19 medical care—such as anticoagulants, vitamins C and D, and zinc—in the emergency departments (EDs), COVID-19 wards, and pharmacy is noted as a health care system insufficiency within the health care facilities. The following extracts from the mortality audit reports support the findings:


*“Appropriate medication out of stock such as Anticoagulation therapy…”*



*“Relevant medication out of stock such as Anticoagulation, vitamin…”*



*“Other treatment out of stock (zinc)… Shortage of medication in the pharmacy.”*



*“No chest X-ray done, no vitamins.”*


Sub-theme 4.2: Improper control and supervision of medical supplies, including essential medicines.

The study noted improper management of medical supplies and equipment, with some of the ordered stock reportedly being received late. The following extracts from the mortality audit reports support the findings:


*“Ordered stock not received on time.”*



*“Treatment ordered received late …led to poor management of patients in the unit.”*


Sub-theme 4.3: Incomprehensive and poorly documented clinical reports.

The current study findings indicate incomprehensible and poor documentation of clinical notes provided in health care services. The incomprehensible and inadequate clinical notes were in the patient files, while other files had nothing written about the health care provided, and in some cases, files were not traceable. The following extracts support these findings:


*“Medical notes missing that led to an incomplete patient assessment record in the brown file.”*



*“Assessments done on the patient but nothing was written…capturing information was not accurate.”*



*“Poor monitoring of the patient, incomplete record of the patient’s assessment.”*


Sub-theme 4.4: Poor staffing of clinical and support staff 

Availability of health care providers and support staff plays a significant role in the patient’s clinical management. However, the current study noted inadequate and inequitable staff distribution as a health care system factor that negatively contributed to COVID-19 mortality. The following extracts support the findings:


*“Staff shortage in the ward... Poor staffing of medical doctors.”*



*“Patient not seen by the doctor at the COVID-19 ward from the day of referral.”*



*“Shortage of staff contributed to inadequate medical care”.*


Sub-theme 4.5: Lack of support from the hospital management and unit managers

Our findings indicate a lack of support from hospital management as a health care system factor contributing to COVID-19 mortality. 


*“Inadequate support from the senior staff members.”*



*“Inadequate support from the staff.”*



*“Inadequate support from senior.”*


Sub-theme 4.6: Poor adherence to COVID-19 standard treatment guidelines

The current study findings indicate poor adherence to the COVID-19 standard treatment guidelines as a health care system factor associated with poor clinical outcomes. The following extracts support the findings:


*“Standard treatment guideline in COVID -19 not adhered to.”*



*“Non-adherence to standard treatment guideline.”*


## 4. Discussion

### 4.1. Sub-Standard Emergency Medical Care Provided

The extract indicates that emergency health care providers could not give timely treatment due to an unprepared emergency trolley and lack of equipment. The poor supply of resources was also reported in several studies in African countries [2,3]. Furthermore, it was reported that these led to delays in treatment and reporting of COVID-19 patients. Hospitals were also overwhelmed with patients and were characterized by high demand for equipment such as ventilators. A similar observation was noted in a study that investigated challenges experienced by health care professionals working in resource-poor intensive care settings in Limpopo Province of South Africa [30]. Consequently, these resulted in low confidence levels, as many patients who did not have hope for survival were taken to the hospital. For several authors, such levels of uncertainty were also attributed to patients’ and health workers’ emotions, perceived stressors, and a severe lack of coping mechanisms during the COVID-19 pandemic [31]. 

Delays in receiving laboratory results due to long turnaround times are interpreted as a health care system error that negatively impacts on the provision of medical intervention. For COVID-19 patients, this often led to failure to diagnose patients on time. Delayed diagnoses are common problems in health care and are typically related to patient, provider, and socioeconomic factors. It was reported that morbidity and mortality from COVID-19 have rapidly risen, especially among black, indigenous, and people of color [32]. Our findings are similar to those reported by Binnicker (2020), who noted high demands for COVI-19 testing laboratory centers [33].

The study suggests that poor monitoring of vital signs and lack of proper investigations, such as X-rays, could have contributed to the delayed provision of appropriate health care for COVID-19 patients. It was observed that some health care providers were fearful and reluctant to interact with COVID-19 patients. Other studies indicate that among the many valid reasons for fear of COVID-19 pandemic is fear of developing an infection, fear of failing to provide adequate care for patients given limited resources, fear of carrying the virus home and infecting family and friends, fear of stigmatization, and many others [34,35]. Fear is not unique to the COVID-19 pandemic; it has been well-described in other disease epidemics such as HIV or SARS [36,37]. Many of these fears were said to be well-founded considering reports of high rates of COVID-19 that were observed among frontline health workers.

Our study findings indicate a delay in the provision of active resuscitation in ED. COVID-19 patients were neither given emergency medical care nor received it early. It is possible that patients overwhelmed the unit resulting in health worker burnout. Burnout has been a major health care issue that has intensified with additional stressors arising from the COVID-19 pandemic. The findings are similar to existing literature which reported common knowledge that frontline health care workers had one of the highest incidences of burnout even before the pandemic [38,39]. Being at the frontline in direct contact with patients suspected or confirmed to have a COVID-19 infection exacerbates this. 

The COVID-19 pandemic revealed the vulnerability of health care systems and how they can rapidly overload the available ICU bed and ventilator capacity. The findings relate to the existing study findings of McCabe et al., who reported that most health care facilities needed to adapt their capacity to COVID-19 demands by providing more trained health care personnel and ventilators [40]. Thus, our study presented a complex phenomenon of high demand in ICU versus the availability of trained personnel who could assist in monitoring patients in need of intensive care. This revealed a weakness in the system which caused failure within facilities. The current findings are similar to those reported by Naidoo (2021), who indicated that COVID-19 had intensified scarce resources in EDs and ICUs [39]. 

### 4.2. Referral System Inefficiencies Contributed to Delay in Access to Health Care Services 

It was observed worldwide during the pandemic that COVID-19 led to the development of hypoxemia in 15–20% of the patients. However, another observation was that after their COVID-19 infection most patients developed hypoxia, a condition where an individual has an alarming low oxygen saturation level of about 50–80% saturation [41]. In contrast, the expected saturation level is ≥95% with the patients not experiencing any difficulties in breathing [42,43]. This phenomenon resulted in patients becoming severely ill, especially when accompanied by a lack of dyspnea and extremely low oxygen saturation levels. Furthermore, these patients were found to be at an unusually high risk of having poor clinical outcomes, which led to patients requiring airflow and ventilator support. Similar observations were made in this study. These are thought to have been aggravated by the fact that our province is very densely populated by poverty and illiteracy, which negatively impacts the patients’ health-seeking practices due to a lack of knowledge. Another contributing factor, in this case, is the observed unavailability of ambulances due to the high demand for emergency medical services within the society, which might have contributed to patients presenting/seeking medical attention late.

In essence, the above observations indicate delays in seeking health care services by several patients who died as a result of COVID-19. These findings align with recent reports from elsewhere [44]. For example, an estimated 41% of U.S. adults reported having delayed or avoided medical care during the pandemic because of concerns about COVID-19, including 12% who reported having avoided urgent or emergency care [44]. Avoidance of both urgent or emergency and routine medical care because of COVID-19 concerns was highly prevalent among unpaid adult-caregivers, respondents with two or more underlying medical conditions, and persons with disabilities. For caregivers who reported caring for adults at increased risk of severe COVID-19, concern about the exposure of care recipients might contribute to care avoidance. Persons with underlying medical conditions that increase their risk of severe COVID-19 are more likely to require care to monitor and treat these conditions, potentially contributing to their more frequent report of avoidance. Moreover, persons at increased risk of severe COVID-19 might have avoided health care facilities because of perceived or actual increased risk of exposure to SARS-CoV-2, particularly at the onset of the pandemic

The current study findings are similar to COVID-19 disease classification in other studies wherein patients can present in mild, moderate and severe-to-critical states [45,46]. Brinton et al. reported mortality amongst bronchial asthma patients who received corticosteroids while presenting with pneumonia [46]. These findings also suggest a need to conduct further COVID-19 investigations on any of those patients who present with asthma or pneumonia-related symptoms. The pandemic brought a great deal of confusion and stress to the functioning of health systems in that medical wards were overwhelmed and overly crowded with moderate, severe, and critical care patients. This brought a high demand for urgent care that could not be achieved due to the shortage of ICU beds and ICU-trained nurses to monitor these patients. 

### 4.3. Advanced Age of Patients with Known and Unknown Comorbidities

Since the COVID-19 pandemic began, there has been insufficient detailed information regarding the causes of death or the contribution of pre-existing health conditions [47]. However, lessons learned from all over the world about the pandemic suggest that the risk of death from COVID-19 is strongly dependent on the patient’s age as well as their other medical conditions [47]. Older patients and those with chronic comorbid pulmonary diseases, such as pneumonia and COPD, were observed during the pandemic to be more susceptible to life-threatening COVID-19 outcomes [32,48,49]. Moreover, age and comorbidity were found to influence most admissions to the intensive care unit (ICU) [50].

The care of patients with endocrine disorders during the COVID-19 pandemic has posed distinctive challenges. Patients with diabetes as a comorbid condition were at an increased risk of developing severe COVID-19 complications. COVID-19 infection precipitated severe manifestations of diabetes, including diabetic ketoacidosis (DKA), hyperosmolar hyperglycemic state (HHS), and severe insulin resistance [47]. Globally, patients with COVID-19 and type 2 diabetes mellitus were more likely to have serious complications that required ICU admissions, a longer length of hospital stay, or resulted in death [51,52,53].

COVID-19 predictions were such that older people with comorbidities would most likely require hospitalization. This group would therefore overwhelm the health care system with demand for critical care. Whilst lockdown level 5 in South Africa aimed at reducing transmission and preparing the health system, extracts indicate that it was mostly the aged population that demanded hospitalization. However, the prognosis was poor on admission. This might indicate poor management of comorbidities common amongst the aging population due to limited access to health care services during the lockdown. Our findings are consistent with other study findings that revealed that patients of higher age and pre-existing chronic health conditions are at an increased risk of experiencing life-threatening COVID-19 clinical outcomes [47,54,55].

### 4.4. Sub-Standard Management of COVID-19 Cases, Medical Records, and Resources 

The pandemic brought a high demand for pharmaceutical services wherein pharmaceutical companies were not coping with this huge demand. Extracts indicate a shortage of essential medicines such as antimicrobials, anticoagulants, vitamins C and D, and zinc. Lockdown restrictions could have negatively impacted the supply chain turnaround time in that, even though the stock was ordered on time, there was never a guarantee of when it would be received. This, therefore, had a negative impact on the provision of proper service delivery to the patients. A similar study by Tan et al. reported the use of analgesia, anti-inflammatories, steroids, anticoagulants, antibiotics, vitamin B, vitamin C and vitamin D and their potential impact on COVID-19 patients [56]. There seem to be limited scholarly studies that report on the shortage of medicines such as antimicrobials, anticoagulants, vitamins C and D, and zinc in rural areas.

Implementing COVID-19 workplace policies, such as social distancing and rotation/shift work, resulted in skeleton staff. This might have negatively impacted the supply chain turnaround time and service delivery. The current study findings are similar to those reported by Bode et al. (2020) who reported challenges with management practices, lack of support, and poor internal communication practices, and human, material, and financial resources during the COVID-19 pandemic, which impacted the quality of patient care [53].

The current study findings indicate gaps in clinical documentation, which negatively impacts the quality of patient care. The present findings align with those reported by Hay, Wilton, Barker, Mortley and Cumerlato (2020), which documented the importance of clinical documentation in Australia [57]. Mutshatshi et al. (2018) report on challenges, such as lack of time to keep clinical records, which affect the quality of patient care [58]. However, there seem to be limited scholarly studies that report on clinical documentation during the COVID-19 era.

There is no doubt that the COVID-19 pandemic caused many challenges in health care facilities and with health care professionals globally. Hence, the findings from the public health facilities in Limpopo Province are similar to the findings in other countries around the world, such as Bangladesh, where health care professionals reported having experienced; (1) an increased workload due to staff shortage brought about by the increasing demand for health care services and the fact that increasingly more health care professionals tested positive; (2) psychological distress; (3) social exclusion and stigmatization; (4) lack of incentives; as well as (5) lack of co-ordination/proper management of their shifts [59,60,61]. Furthermore, health care professionals reported that during the second wave of the pandemic they had difficulty coping with all these challenges due to situational and organizational factors [59]. This is thought to have resulted from the fact that, when the second wave came, increasingly more people tested positive, including many health care professionals.

A wide range of work-related stressors in health care facilities associated with the COVID-19 pandemic was experienced. This was brought by the enormous number of patients that constrained hospitals’ capacities resulting in additional workplace-related stress, especially among emergency health care providers. Additionally, working hard during emergencies and in stressful working conditions is often associated with sleep deprivation amongst health care professionals with an increased risk of burnout [62]. Similar to burnout, an excessive level of stress in health care is a serious factor that has the potential to affect health care professionals’ working environments and, therefore, compromise their performance, especially during an emergency [63]. Therefore, it is necessary that adequate support to address the difficulties faced by health care professionals is provided by management for an overall improved health outcome during the pandemic [63].

In terms of the COVID-19 pandemic, we noted insufficient scholarly work addressing the factors associated with compliance with treatment guidelines [31]. However, the observations from clinical practice are that the overall issue of adherence to prescribed standard treatment guidelines is highly dependent on the clinicians’ ability to read and understand the guidelines. It is possible that some health care professionals might have felt that the COVID-19 treatment guidelines were not straightforward and were confusing, and this resulted in them simply applying their clinical knowledge to manage the situations they were confronted with. To improve health care services and the effectiveness of emergency medical services provided during the pandemic, there is, therefore, a need for health care professionals to be taught and closely monitored when implementing new treatment guidelines.

## 5. Conclusions

The clinical audit reports present a unique exploration of the health care system and patient-related factors regarding the COVID-19 mortality rate in Limpopo Province. In particular, this study presents novel findings regarding the impact of health care system-related and patient-related factors on the COVID-19 mortality rate. Clinical audit reports indicated health care system-related and patient-related factors which brought many uncertainties regarding the management of COVID-19 patients in health care settings. Some of these uncertainties resulted in sub-standard clinical management, which provided an opportunity for a rise in lawsuits and poor COVID-19 inpatients outcomes. Sub-standard management of COVID-19 is compounded by many factors, such as poor adherence to COVID-19 treatment guidelines, fear of health care providers, market competition for and shortage of medical supplies/equipment, and infected health care providers. We interpreted these factors as the health care system-related and patient-related factors that resulted in poor outcomes amongst COVID-19 inpatients. COVID-19 progression and lack of knowledge among health care providers resulted in most patients seeking medical assistance late. The global shortage of medical supplies and equipment negatively impacted patients’ management. There is a need to routinely conduct clinical audits to improve the quality of health care services by identifying patient and health care system challenges to develop appropriate recommendations. We recommend a review of health promotion, risk communication, and community engagement strategies to improve patients’ health-seeking behaviors. There is also a need to review and expand the scope of practice among health care providers during epidemics and pandemics to include aspects such as task-shifting to address staff burnout and shortages. The audit has also revealed poor co-ordination and management of health resources. We recommend drills to prepare health care providers for disaster management.

## Figures and Tables

**Table 1 healthcare-10-01338-t001:** Hospital audit team.

Designation	Role
Physician/family physician	Clinical audit leader
Medical practitioner (COVID-19 ward)	Clinical notes audit
Professional nurse(COVID-19 ward)	Clinical notes audit
Operational manager(COVID-19 ward)	Clinical notes audit
Quality assurance	Clinical notes audit
Infection prevention and control (IPC)	Compilation of mortality line list
Patient records	Patient record retrieval
Health information	Capturing of data
Data capturer	Capturing of data

**Table 2 healthcare-10-01338-t002:** Themes and sub-theme on the health system and patient-related factors associated with COVID-19 mortality among hospitalized patients.

**Themes**	**Sub-Themes (Frequency)**
Sub-standard emergency medical care provided	1.1. Delayed triage and clinical assessment (*n* = 47). 1.2. Delayed laboratory results turnaround time (*n* = 86). 1.3. Insufficient clinical and radiological investigations (*n* = 90). 1.4. Insufficient active resuscitation of severe COVID-19 cases (*n* = 138). 1.5. Inadequate monitoring of COVID-19 in the emergency unit (*n* = 21).
2. Referral system inefficiencies contributed to delay in access to health care services	2.1. Severe and critical cases presenting with hypoxia on admission (*n* = 92). 2.2. Patients presenting with respiratory complications on admission (*n* = 389).
3. Advanced age of patients with known and unknown comorbidities	3.1. Advanced age and respiratory complications (*n* = 389). 3.2. Patients with diabetes mellitus, diabetic ketoacidosis (DKA), hyperglycemia, or hypoglycemia (*n* = 139). 3.3. Advanced aged patients presented with comorbidities and severe stage of COVID-19 (*n* = 217).
4. Sub-standard management of COVID-19 cases, medical records, and resources.	4.1. Out of stock medical supplies in the wards and pharmacy (*n* = 92). 4.2. Improper control and supervision of medical supplies, including essential medicines (*n* = 52). 4.3. Incomprehensive and poorly documented clinical reports (*n* = 80). 4.4. Poor staffing of clinical and support staff (*n* = 50). 4.5. Lack of support to health care professionals by hospital management and unit managers (*n* = 107). 4.6. Poor adherence to COVID-19 standard treatment guidelines (*n* = 50).

## Data Availability

Not applicable.

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
