# Peer review of "Health System- and Patient-Related Factors Associated with COVID-19 Mortality among Hospitalized Patients in Limpopo Province of South Africa’s Public Hospitals"

_healthcare, 2022, doi:10.3390/healthcare10071338_

Round 1
Reviewer 1 Report
The authors present a study that "focuses on analyzing COVID-19 mortality data to discover the health care system errors related to the COVID-19 mortality rate."
1. It is unclear to me whether this is the aim of the study. If this is the aim, the methods do not reflect this aim completely. It is essential that the aim of the study is clearly stated.
2. Analysis of mortality data will result in a quantitative process, rather than a qualitative one. The authors have not presented an aim that captures the qualitative methods employed.
3. The authors present a thematic data analysis of the reviewed 770 COVID-19 patients` records. Essentially this means that a content analysis was performed, yet not described in the methods
4. The authors describe the clinical audit process in which open-ended questions guided discussions on “What happened? Error analysis. Describe the sequence of events leading to adverse outcome”. But it is unclear who facilitated these discussions. The researchers or the audit team? How were the data captured? By whom? If the researchers were not part of the process, does it mean that they analysed the clinical audit reports? These need to be clearly described
5. Data analysis: Who did what? How many people were involved in the coding? Where codes independently developed and reconciled? How?
6. It is unclear if the authors were among the clinical audit team. If they were, how does this influence the interpretation of the findings. A note on reflexivity is essential.
7. The study does methods does not follow the Consolidated criteria for reporting qualitative research (COREQ). I recommend that the authors submit a copy of the completed COREQ checklist as an attachment, as recommended by the EQUATOR network. (https://www.equator-network.org/reporting-guidelines/coreq/). This would both help the authors adequately describe their methods, as well as allow for a standardized review of their work.
Author Response
The authors present a study that "focuses on analyzing COVID-19 mortality data to discover the health care system errors related to the COVID-19 mortality rate."
- It is unclear to me whether this is the aim of the study. If this is the aim, the methods do not reflect this aim completely. It is essential that the aim of the study is clearly stated.
- The current study sought to determine the health care system and patient-related factors by reviewing the COVID-19 inpatient mortality audit narration reports.
- Analysis of mortality data will result in a quantitative process, rather than a qualitative one. The authors have not presented an aim that captures the qualitative methods employed.
- The authors tried to put more clarification on the aim of the study “The current study sought to determine the health care system and patient-related factors by reviewing the COVID-19 inpatient mortality audit narration reports”
- The authors present a thematic data analysis of the reviewed 770 COVID-19 patients` records. Essentially this means that a content analysis was performed, yet not described in the methods
- The authors changed it to content analysis. “Content analysis was conducted using a systematic coding and categorizing approach to explore data from 770 records within all 41 public hospitals to identify themes regarding health system and patient-related factors associated with COVID-19 mortality”
- The authors describe the clinical audit process in which open-ended questions guided discussions on “Please explain what could have caused the mortality? Error analysis. Describe the sequence of events leading to adverse outcome”. But it is unclear who facilitated these discussions. The researchers or the audit team? How were the data captured? By whom? If the researchers were not part of the process, does it mean that they analyzed the clinical audit reports? These need to be clearly described
- The authors added this to give more clarity “The tool was administered and completed by selected facilitators identified by the Provincial Case Management and Audit team who are not part of the authorship. The audit was conducted in a participatory manner, in a group setting with representatives from the hospital consisting of the hospital clinical manager, Infection control nurse and a DATCOV data capture owing to their familiarity and knowledge of the context of the demise of COVID-19 inpatients”. The Provincial Case Management and Audit Team captured all the narrations about the mortality audit reports in the excel spreadsheet.
- Under study design authors mentioned that “The study adopted a retrospective qualitative design to analyze all deaths that resulted from hospitalized COVID-19 patients between the first (16 March 2020 - 31 October 2020) and second (01 November 2020 - 31 March 2021) pandemic waves. Secondary data was obtained from the COVID-19 mortality audit narration reports kept by the Limpopo Province Department of Health (LDoH).”
- Data analysis: Who did what? How many people were involved in the coding? Where codes independently developed and reconciled? How?
- The adoption of content analysis enabled authors to determine the trends and patterns of words used, their frequency and their relationship (Vaismoradi et al., 2013). The next step was to conduct open codimg and create themes and group the codes in higher order headings, alignining with the objectives of the study themes and sub-themes were formu-lated as emerged. Data was reassembled in new ways, with connections between codes and similar codes grouped into broader themes and subthemes. Two authors(MOM & L) completed the data analysis by immersing themselves with the data, to obtain a sense of whole transcripts before generating the themes and sub-themes. To validate the themes and sub-themes, four authors (PMM, LNM, MOM & L) discussed and verified the generated themes and sub-themes, which was confirmed by the first author(MSS). Five re-searchers and decide on the analysis
- It is unclear if the authors were among the clinical audit team. If they were, how does this influence the interpretation of the findings. A note on reflexivity is essential.
- Authors didn’t form part of the audit team: “The study adopted a retrospective qualitative design to analyze all deaths that resulted from hospitalized COVID-19 patients between the first (16 March 2020 - 31 October 2020) and second (01 November 2020 - 31 March 2021) pandemic waves. Secondary data was obtained from the COVID-19 mortality audit reports kept by the Limpopo Province Department of Health (LDoH).”
- The study does methods does not follow the consolidated criteria for reporting qualitative research (COREQ). I recommend that the authors submit a copy of the completed COREQ checklist as an attachment, as recommended by the EQUATOR network. (https://www.equator-network.org/reporting-guidelines/coreq/). This would both help the authors adequately describe their methods, as well as allow for a standardized review of their work.
- The study followed COREQ see attached.

Reviewer 2 Report
General comments
=============
Thank you for letting me peer-review your work! This paper is an interesting study for the health system and patient-related factors associated COVID-19 mortality.
Specific comments
=============
Major comments
---------------------
1. Please clarify how to set the themes and sub-themes.
2. It is better to separate the result, discussion, and limitation. It was unclear what part was the author’s interpretation and what part was the audit team discussion.
3. Please clarify how many records match each theme.
4. If available, please add patient-related data. For example, sub-themes 3.1 was related to the patients’ advanced age. However, there was no age data. If available, please clarify how to confirm COVID-19 infection; PCR, Antibody, or clinically.
5. Please discuss these themes were unique for COVID-19 mortality or also common for non-COVID-19 mortality.
Minor comments
---------------------
6. Please arrange in descending order of the underlying conditions in figure 1.
7. Please explain the abbreviation of “PUI”.
8. Please change Table “1” to Table “2” in lines 167 & 171.
Author Response
Thank you for letting me peer-review your work! This paper is an interesting study for the health system and patient-related factors associated COVID-19 mortality.
Major comments
---------------------
- Please clarify how to set the themes and sub-themes.
- Authors modified the data analysis section as “ Content analysis was conducted using a systematic coding and categorising approach to explore data from 770 records within all 41 public hospitals to identify themes regarding health system and patient-related factors associated with COVID-19 mortality. The adoption of content analysis enabled authors to determine the trends and patterns of words used, their frequency and their relationship [28]. The next step was to conduct open codimg and create themes and group the codes in higher order headings, alignining with the objectives of the study themes and sub-themes were formulated as emerged. Data was reassembled in new ways, with connections between codes and similar codes grouped into broader themes and subthemes. Two authors(MOM & L) completed the data analysis by immersing themselves with the data, to obtain a sense of whole transcripts before generating the themes and sub-themes. To validate the themes and sub-themes, four authors (PMM, LNM, MOM & L) discussed and verified the generated themes and sub-themes, which was confirmed by the first author(MSS). Five researchers and decide on the analysis. Throughout the coding process, regular meetings were held by authors to discuss and finalise themes and sub-themes. This was done to improve data trustworthiness using a virtual platform (Google meet)”
- It is better to separate the result, discussion, and limitation. It was unclear what part was the author’s interpretation and what part was the audit team discussion.
- The authors separated the results (lines 195 to 361) and the discussion (lines 362-542) in the manuscript report.
- Please clarify how many records match each theme.
- The authors added the frequency for each sub-theme see Table 1
- If available, please add patient-related data. For example, sub-themes 3.1 was related to the patients’ advanced age. However, there was no age data. If available, please clarify how to confirm COVID-19 infection; PCR, Antibody, or clinically.
- The authors added that under research design. “The study included all patient clinical records of laboratory-confirmed COVID-19 cases and excluded non-laboratory-confirmed COVID-19 cases”.
- Please discuss these themes were unique for COVID-19 mortality or also common for non-COVID-19 mortality.
- The authors showed that “The most common types of preventable health system errors were those in diagnosis, delayed diagnosis, medication errors, and deficient medical devices [9]. Medication errors were reported to be common among COVID-19 patients in a large tertiary care setting in the Kingdom of Saudi Arabia. They were caused by dosing errors and errors in drug frequency, which were primarily attributed to physicians, which may indicate burnout or stress among them [10]” . Line 70-75
- The authors showed that “Sub-standard management of COVID-19 is compounded by many factors, such as poor adherence to COVID-19 treatment guidelines, fear of healthcare providers, market competition for and shortage of medical supplies/equipment, and infected healthcare providers. We interpreted these factors as the health care system and patient-related factors that resulted in poor outcomes amongst COVID-19 inpatients. COVID-19 progression and lack of knowledge among health care providers resulted in most patients seeking medical assistance late”. Line 552-558
Minor comments
---------------------
- Please arrange in descending order of the underlying conditions in figure 1.
- Figures were removed as it was reported before in other articles.
- Please explain the abbreviation of “PUI”.
- The abbreviation was added.
- Please change Table “1” to Table “2” in lines 167 & 171.
- Table 1 in lines 197-198 was changed to Table 2
Round 2
Reviewer 1 Report
My concerns have been addressed. Minor typos need to be checked. For example, line 164 uses 'codimg' instead if 'coding'.
Author Response
Thank you for all the comments and recommendations. The opinions were really helpful to eliminate misunderstandings and several major flows and to make the manuscript more accurate and valuable.
The authors checked all possible minor typos and made corrections to the document.
Reviewer 2 Report
General comments
=============
Thank you for letting me peer-review your work! This paper is an interesting study for the health system and patient-related factors associated COVID-19 mortality. Almost all responses from author were reasonable.
Author Response
Thank you for all the comments and recommendations. The opinions were really helpful to eliminate misunderstandings and several major flows and to make the manuscript more accurate and valuable